# A Pilot Study Investigating the Relationship Between Choroidal Thickness and Choroidal Vascular Index and Coronary Artery Ectasia

**DOI:** 10.3390/diagnostics15030286

**Published:** 2025-01-26

**Authors:** Dogukan Comerter, Tufan Cinar

**Affiliations:** 1Department of Ophthalmology, Sultan Abdülhamid Han Training and Research Hospital, University of Health Sciences, 34668 Istanbul, Turkey; 2Department of Cardiology, Sultan Abdülhamid Han Training and Research Hospital, University of Health Sciences, 34668 Istanbul, Turkey; drtufancinar@gmail.com; 3Department of Internal Medicine, University of Maryland Midtown Campus, Baltimore, MD 21201, USA

**Keywords:** coronary artery ectasia, cardiovascular diseases, choroidal thickness, choroidal vascular index, optical coherence tomography

## Abstract

**Objective**: The objective of this study was to compare changes in choroidal thickness (ChT) and choroidal vascular index (CVI) between patients with coronary artery ectasia (CAE) and healthy individuals. **Methods**: This study included 34 patients with CAE and 40 age-matched healthy subjects with normal coronary arteries. Measurements of ChT and CVI were taken using spectral-domain optical coherence tomography, employing the binarization method for CVI calculation. Additional parameters, including central macula thickness (CMT), retinal nerve fiber layer (RNFL), and ganglion cell layer (GCC) thickness, were also documented. **Results**: The results indicated no significant differences in either subfoveal ChT or in ChT at 1500 µm both nasal and temporal to the fovea. However, significant differences were noted in ChT at the 500 µm nasal and temporal areas. The CVI was found to be significantly lower in the CAE group compared to the healthy controls. Furthermore, this study noted a significant difference in GCC thickness between the two groups, while no significant differences were observed in CMT and RNFL measurements. **Conclusions**: The findings suggest that patients with CAE exhibit decreased ChT and CVI in comparison to healthy controls. This highlights the potential role of ChT and CVI as important markers of disease in coronary artery ectasia, offering valuable insights into systemic cardiovascular health.

## 1. Introduction

Coronary artery ectasia (CAE) is an uncommon but significant condition that pertains to coronary artery disease (CAD), characterized by an abnormal and considerable dilatation of a coronary artery. This dilatation is quantified as extending beyond one-third of the artery’s overall length and achieving a diameter that is at least 1.5 times greater than that of a comparable normal artery located nearby [1]. CAE can present in two forms: focal, where a specific segment of the artery is affected, or diffuse, where the ailment extends along a broader section of the artery. Of note, the right coronary artery is the most commonly implicated vessel. Patients diagnosed with CAE face an elevated risk of experiencing acute coronary syndrome, a serious condition that can lead to myocardial infarction [2]. Contributing factors to the development of CAE include hypertension (HT), smoking habits, and the presence of atherosclerotic heart diseases (AHDS), all of which further compromise cardiovascular health [3,4].

The eye functions as a vital organ that not only plays a key role in vision but also serves as an essential indicator of systemic health. The microvascular effects of various systemic diseases can be observed directly within the eye, allowing medical professionals to garner insights that may not be as readily evident through other diagnostic means. By assessing specific visual parameters, health practitioners can gain predictive and prognostic insights crucial for managing and mitigating systemic complications associated with HT and diabetes mellitus (DM), as well as cardiovascular and cerebrovascular diseases [5]. The advent of advanced imaging technologies, particularly spectral-domain optical coherence tomography (SD-OCT), has revolutionized the field by enabling detailed visualization and precise measurement of retinal structures and choroidal thickness (ChT). Moreover, newer imaging techniques such as enhanced-depth imaging-mode OCT (EDI-OCT) and optical coherence tomography–angiography (OCT-A) provide superior visualization of the choroid and its associated vascular plexuses, surpassing the capabilities of traditional imaging methods [6].

Numerous studies have indicated that alterations in the microvascular structure of the choroid can serve as early indicators of systemic diseases that adversely impact blood vessels throughout the body. This evidence underscores the importance of understanding the intricate relationship between choroidal changes and cardiovascular diseases (CVDs) in a clinical setting [5,7,8]. Current measurement methodologies, including ChT and choroidal vascularity index (CVI), stand as prominent noninvasive techniques available for analysis. Earlier research has delved into CVI in individuals suffering from coronary artery disease (CAD), revealing significant and clinically relevant differences in this patient population [9]. The ChT is significantly diminished in hypertensive individuals when compared to their normotensive counterparts [10]. Furthermore, those with pseudoexfoliation syndrome exhibit notably reduced ChT relative to healthy controls [11]. However, the specific relationship between CAE and changes in the choroid has not yet been thoroughly investigated. Given that the early identification of vascular changes in patients with CAE could play a crucial role in influencing their overall prognosis, this study aims to be the first to carry out a comprehensive comparison of variations in ChT and CVI between individuals diagnosed with CAE and healthy control subjects. By shedding light on these connections, we hope to enhance the understanding of CAE’s systemic implications and ultimately contribute to improved patient care and outcomes.

## 2. Materials and Methods

### 2.1. Study Participants

A prospective, single-center study was meticulously conducted from April 2021 to December 2023 at Sultan Abdulhamid Training and Research Hospital’s Ophthalmology Department in collaboration with the Cardiology Department. Each patient enrolled in this study underwent a thorough assessment by a highly experienced cardiologist. This assessment included compiling an exhaustive cardiac history alongside a thorough physical examination to identify any underlying cardiovascular issues. Following this initial evaluation, patients were subjected to carefully designed cardiovascular stress testing, which could either be exercise-based or pharmacological. Such testing was aimed at identifying patients who exhibited moderate to severe ischemia as evident on imaging results and who subsequently underwent coronary angiography (CAG). Additionally, those assessed as high risk based on their responses during the exercise testing were referred for CAG.

To uphold the scientific rigor of our findings, we established stringent exclusion criteria. Patients with any ocular diseases—including but not limited to glaucoma, uveitis, or various retinal disorders like diabetic retinopathy, macular edema, and age-related macular degeneration—were excluded from this study. To ensure the integrity of our study, individuals exhibiting a refractive error greater than ±3.0 D, along with those having long (≥25 mm) or short (≤22 mm) axial lengths, were also excluded from this study. Additionally, individuals with a history of previous retinal treatments such as laser photocoagulation or intravitreal injections, along with those who had undergone any intraocular surgeries other than uncomplicated phacoemulsification, were also not considered for participation. On the cardiovascular front, patients presenting severe valvular obstruction or regurgitation as determined by echocardiography or those with histories of percutaneous coronary intervention (PCI) or coronary artery bypass grafting (CABG) were excluded (Figure 1). Ultimately, our study encompassed a total of 34 patients diagnosed with CAE. In parallel, we formed a control group of 40 carefully selected age-matched individuals with normal coronary arteries, which served as a point of comparison. The study protocol was subjected to rigorous scrutiny and received formal approval from the Local Ethics Committee, adhering strictly to the ethical standards outlined in the Declaration of Helsinki (HNEAH-KAEK-2023/208). Importantly, we ensured that informed consent was meticulously obtained from every participant prior to their involvement in this study, highlighting our commitment to ethical research practices.

### 2.2. Study Protocol and Procedure

The CAG was performed using Judkins’s technique through either radial or femoral access points. In our study, CAE was classified as an observable expansion of the coronary artery exhibiting a diameter of 1.5 times or more compared to a standard, healthy segment of the artery. A normal segment was distinctly characterized by the absence of CAE as well as coronary artery stenosis, thus ensuring that cases displaying associated coronary stenosis were excluded from our analysis. Patients identified with CAE were further categorized into two distinctive groups: focal CAE, where the extension is localized, and diffuse CAE, where the expansion is spread over a more extensive area. Comprehensive demographic information was meticulously gathered, accompanied by an analysis of contributing factors, such as HT, DM, hyperlipidemia (HL), and smoking habits, that may influence the presence of CAE.

In the realm of ophthalmic evaluations and imaging, a single physician performed all assessments and SD-OCT imaging in a strictly blinded manner, ensuring objectivity and minimizing bias. All measurements were conducted during the early morning hours, specifically between 9:00 AM and 11:00 AM, following pupil dilation to minimize the effects of diurnal rhythm variations. Pupil dilation was achieved through the careful instillation of topical tropicamide 1%, ensuring accuracy and reliability in our results. The evaluations encompassed several critical components, including measuring best-corrected visual acuity (BCVA) through the use of a Snellen chart, conducting anterior segment examinations utilizing slit-lamp biomicroscopy, measuring intraocular pressure (IOP) for glaucoma assessment, and performing thorough funduscopic examinations. Remarkably, all participants exhibited a BCVA of 20/20 and demonstrated IOP levels below the established threshold of 21 mmHg. OCT images were obtained using the advanced Spectralis OCT device (Software version 6.16.8), which features cutting-edge eye-tracking dual-beam technology provided by Heidelberg Engineering GmbH in Heidelberg, Germany. A randomly selected eye of each participant served as the study eye; if this eye did not meet the inclusion criteria, assessments were shifted to the fellow eye. Additionally, we meticulously recorded measurements of central macula thickness (CMT), retinal nerve fiber layer (RNFL), and ganglion cell layer (GCC) to evaluate the health of retinal structures.

For the precise measurements of ChT, we utilized a high-fidelity precision caliper supplied by EDI-OCT, a renowned tool in ocular diagnostic imaging. This technique enabled a detailed assessment of the perpendicular distance between the hyperreflective outer border of the retinal pigment epithelium and the Bruch membrane layer. Measurements were carefully taken at five critical points along a horizontal scan line, directly at the subfoveal region, as well as at points 500 μm and 1500 μm both temporally and nasally from the fovea (Figure 2). To ensure the highest level of accuracy and consistency, if any single measurement demonstrated a deviation exceeding 10%, we promptly conducted a second round of measurements to verify the results and maintain data integrity.

Within our analysis of the choroidal area (CA), we conducted comprehensive measurements over a total area of 3000 μm, including a margin of 1500 μm extending both nasally and temporally from the center of the fovea. This meticulous vertical measurement extended from the retinal pigment epithelium (RPE) down to the choroidoscleral border, allowing for an in-depth evaluation of the choroidal vascular structure. We identified the edges of the CA using the ImageJ Region of Interest (ROI) Manager (Version 1.50a; National Institutes of Health, Bethesda, MD, USA), a sophisticated software tool designed for precise image analysis. Following this step, we applied binarization through the advanced Niblack auto-local threshold method to enhance image clarity by categorizing pixel intensity. In the processed binarized image, darker pixel areas signified vascular channels, referred to as the luminal area (LA), while lighter pixel areas indicated the stroma of the choroid, termed the stromal area (SA). The CVI was subsequently calculated as the ratio of LA to the total CA, providing invaluable insights into the vascular health and structural integrity within the choroid (Figure 3).

All manual measurements were executed by a single physician to guarantee consistency and to minimize inter-observer variability. Notably, this physician conducted all measurements while blinded to the group assignments during the measurement process, effectively eliminating any potential bias. Any measurements revealing discrepancies greater than 10% were rigorously excluded from the final analysis, upholding the integrity and credibility of this study’s findings. This meticulous approach not only enhances the reliability of our data but also significantly contributes to our understanding of choroidal dynamics in various ocular conditions, providing a crucial foundation for future research and clinical insights.

### 2.3. Statistical Analysis

Statistical analyses were meticulously performed using SPSS 22.0 software, developed by SPSS Inc. based in Chicago, IL, USA, to ensure precise and accurate evaluation of the data. The power analysis demonstrated that involving at least 33 patients and healthy individuals was essential to achieve a statistically significant difference of 0.03 between the groups, ensuring a robust power of 0.80. Initially, we assessed the distribution of the dataset using the Shapiro–Wilks test, which allowed us to determine whether the data followed a normal distribution. Continuous variables were clearly expressed as mean ± standard deviation (SD) to provide a comprehensive overview of the central tendency and variability. In contrast, categorical variables were presented as frequency counts and percentages to illustrate their distribution within the sample population. To compare continuous variables between the two groups, we employed the independent samples *t*-test for normally distributed data, while the Mann–Whitney U test was utilized for data that did not meet the assumptions of normality. For categorical variables, we used the chi-square (x^2^) test, which enabled us to assess the association between variables across the two groups effectively. Additionally, we conducted correlation analyses using either the Pearson correlation test for linear relationships or the Spearman correlation test for nonparametric data, allowing us to explore potential associations between variables more thoroughly. Throughout all analyses, we adhered to a significance level of *p* less than 0.05, which indicated statistical significance and reinforced the validity of our findings. By implementing these rigorous analytical methods, we aimed to provide robust and trustworthy conclusions based on our data.

## 3. Results

The study encompassed a total of 74 participants, with 34 individuals specifically diagnosed with CAE and a control group comprising 40 healthy participants, both presenting with a closely matched mean age—61.1 ± 8.8 years for the CAE patients and 61.2 ± 7.9 years for the controls. A comprehensive overview of the demographic and clinical characteristics of both groups is provided in Table 1, where it becomes evident that individuals with CAE exhibited a significantly higher likelihood of comorbid conditions such as HT, DM, and HL.

In terms of ocular metrics, the mean subfoveal ChT was measured to be distinctly lower in the CAE cohort, registering at 297.88 ± 73.2 µm, compared to the control group’s average of 336.3 ± 94.4 µm. This difference in choroidal thickness is effectively illustrated in Figure 1, which depicts measurements taken at five different locations along the retina, underscoring the variations in structural health between the two groups. A significant difference was observed in ChT at 500 µm nasal and temporal area. The choroid was significantly thinner at these locations in the CAE group (*p* < 0.05) (Table 2).

Furthermore, the evaluation utilized advanced OCT techniques to assess several critical parameters. The findings showed that the GCC, RNFL, and CMT values in patients with CAE were all significantly reduced when juxtaposed with those of the healthy controls. Particularly noteworthy was the significant difference observed in GCC thickness; however, variations in CMT and RNFL did not reach statistical significance. Importantly, the CVI was found to be significantly lower in the CAE group compared to the control participants, with a *p*-value of 0.0001, as detailed in Table 3. Despite these differences, this study did not find a significant correlation between subfoveal ChT and CVI, suggesting that alterations in these metrics may be independent of one another.

Additionally, the research explored the effects of systemic risk factors, including HT, DM, DM, and smoking habits, on ChT and CVI specifically within the CAE group. The evaluation revealed no significant differences related to these systemic factors, as summarized in Table 4.

## 4. Discussion

The choroidal circulation is fascinatingly complex and boasts one of the highest rates of blood flow in the entire human body [12]. This robust network is not merely a passive conduit; it actively supplies oxygen and essential nutrients to the retinal layers, which stretch from the retinal pigment epithelium all the way to the inner nuclear layer. Importantly, the choroid also assists in removing metabolic waste products generated by various retinal cells, thereby playing a critical role in maintaining the retina’s overall health and functionality [13]. Additionally, the choroidal circulation contributes to the regulation of temperature in the eye, ensuring optimal conditions for visual processing. Given its multifaceted functions, it is evident that maintaining a healthy choroidal vasculature is essential for the retina to operate effectively. The choroidal arteries possess a distinctive architecture, particularly in the choriocapillaris, which is densely packed and uniquely structured to facilitate efficient nutrient exchange [14]. This morphology suggests that the choroid can be evaluated as an end-organ, making it a valuable indicator for assessing cardiovascular health, particularly in risk stratification for patients with potential CAD [8].

Moreover, various systemic diseases can significantly influence ChT, including HT, DM, carotid artery stenosis, heart failure, and HL [5,15,16]. For instance, acute instances of hypertension have been shown to result in an increase in subfoveal ChT, suggesting an adaptive response to elevated blood pressure [17]. Conversely, chronic hypertension may lead to a detrimental reduction in ChT, indicating potential long-term harm to the choroidal and retinal structures [7]. Prior studies have predominantly focused on ChT in patients suffering from CAD, with these investigations revealing that ChT was notably thinner in CAD patients than in a control group [18].

In our comprehensive study, we took a novel approach by examining both ChT and CVI in a specific patient cohort suffering from CAE. Our meticulous analysis demonstrated a significant decrease in ChT across all five measured locations—subfoveal, nasal 500 µm, nasal 1500 µm, temporal 500 µm, and temporal 1500 µm—in the CAE group. Notably, reductions at both the 500 µm nasal and temporal regions reached statistical significance, reinforcing the implications of impaired choroidal health in this context. Additionally, we observed marked decreases in CMT, RNFL, and GCC thickness in the CAE group, with particularly noteworthy reductions in GCC thickness.

Furthermore, a recent study conducted by Won et al. [9] aimed to elucidate the relationship between subfoveal ChT and CVI in patients with suspected CAD. Although this study noted no significant differences in subfoveal ChT and CVI between patients with and without CAD, it did reveal that CVI levels were significantly lower in individuals presenting with triple-vessel disease compared to those without CAD or those with one–two vessel involvement. The insights from our current study affirm that CVI values were significantly lower in the CAE group when compared to a control group, emphasizing the importance of assessing choroidal parameters for insights into cardiovascular health and disease.

The intricate relationship between the choroid and CAD offers a compelling area for investigation, particularly considering the overlapping pathophysiological mechanisms involved. CAD is primarily characterized by atherosclerotic changes in medium to large blood vessels. In contrast, the diameter of choroidal vessels is more analogous to that of coronary microvessels, which could suggest functional parallels between these two vascular systems [19,20]. Microvascular CAD emerges when there is evidence of coronary ischemia without the presence of significant obstructive atherosclerosis typically seen in CAD patients. Additionally, the clinical manifestations of CAE often coexist with ASHD, presenting a complex intersection that warrants further exploration. Many patients diagnosed with CAE also demonstrate atherosclerotic changes, which raises critical questions about potential underlying common pathophysiological processes. It remains to be clarified whether CAE is simply a variant of ASHD or if it embodies a distinct entity altogether. Notably, CAE can occur in isolation, lacking significant stenosis, thereby increasing the likelihood of turbulent hemodynamic flow. This turbulence can raise the risk of ischemia not only in the coronary system but potentially in other vascular beds as well [21,22].

Our comprehensive study reveals that localized thinning of the choroid, accompanied by a low CVI among patients with CAE, may be a direct consequence of disrupted microvascular perfusion both in the eye and in other organs. Moreover, we expanded our analysis to assess the influence of systemic risk factors on ChT and CVI in this patient population. A significant number of researchers have established that the choroid tends to experience thinning in individuals with HT. Furthermore, the existing literature suggests that variations in the choroidal structure can be significantly linked to the duration of DM. Therefore, patients with DM may exhibit fluctuations in ChT determined by multiple factors, including the length of diabetes, the presence and severity of retinopathy or macular edema, the stage of retinopathy, and the extent of treatment being received. In addition, Wong et al. [9] posited that subfoveal ChT is generally thicker in cases of HL, attributing this increase to factors such as lipid accumulation within the suprachoroid and the hypertrophy of vascular smooth muscle cells. However, intriguingly, our study found no significant differences in ChT or CVI within the CAE group regarding the presence or absence of common systemic risk factors such as HT, HL, or DM. This nuanced understanding of the vascular interplay and its implications for patient management is critical for both clinical practice and future research initiatives.

This comprehensive examination highlights the substantial impact of CAE on ocular health and poses essential considerations for future clinical assessments and interventions. The findings advocate for deeper inquiry into the interplay between systemic health conditions and ocular pathologies, thereby fostering a more thorough understanding of CAE and its implications on visual health.

## 5. Limitations of the Study

When interpreting the findings of this study, it is essential to consider several key limitations that may impact the overall conclusions drawn. Firstly, while the sample size was relatively small, it is noteworthy that we enrolled all consecutive patients who presented at the center during the study period. This approach aimed to reduce selection bias and enhance the reliability of our findings. Moreover, the statistical power of the study was deemed adequate for detecting significant differences, which strengthens the validity of our results despite the small sample. Secondly, our research was conducted at a single medical center within a specific geographical location. This limitation raises questions about the generalizability of our findings to broader populations. Therefore, it is crucial that future studies involve larger sample sizes and multiple centers across diverse geographical areas to confirm and expand upon our outcomes. Carotid intima thickness and the carotid intima/media index are widely recognized as significant risk factors for coronary disease. Nevertheless, it is important to note that our study did not include any data regarding carotid intima thickness or the carotid intima/media index. We acknowledged that various factors, including axial length, can significantly impact ChT and CVI. Furthermore, while we found statistically significant differences in ChT and CVI between patients with CAE and age-matched controls, an important limitation of our study was the inability to collect long-term follow-up data. Longitudinal data would provide invaluable insights into the progression of choroidal changes and their clinical significance over time.

## 6. Conclusions

Our analysis revealed that patients with CAE exhibited notable discrepancies in choroidal vascular parameters compared to their age-matched counterparts. The findings suggest that alterations in ChT and CVI may be indicative of underlying pathophysiological processes associated with CAE. Specifically, these parameters could potentially serve as significant disease markers, providing critical information regarding systemic cardiovascular health and susceptibility to diseases affecting the outer retina and RPE.

## 7. Future Directions

We want to underscore that while our study may not yet translate into immediate clinical applications, it highlights a noteworthy association between ChT and CVI and CAE. Furthermore, by establishing this crucial connection, we pave the way for further research that will delve deeper into these associations and uncover the clinical implications that can emerge from a better understanding of these vascular changes.

## Figures and Tables

**Figure 1 diagnostics-15-00286-f001:**
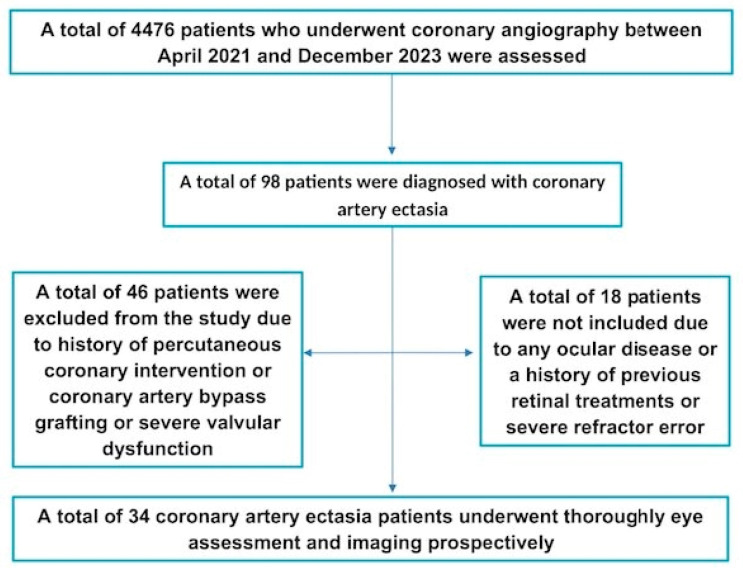
The flowchart of study participants shows the number of patients who met inclusion or exclusion criteria after coronary angiography.

**Figure 2 diagnostics-15-00286-f002:**
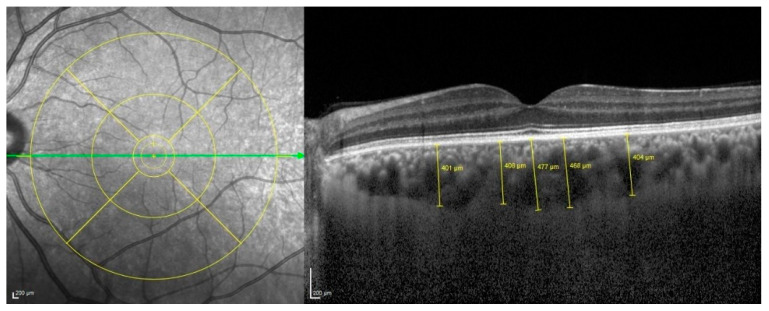
These pictures show choroidal thicknesses obtained at 5 different points (subfoveal, nasal 500 µm, temporal 500 µm, nasal 1500 µm, temporal 1500 µm).

**Figure 3 diagnostics-15-00286-f003:**
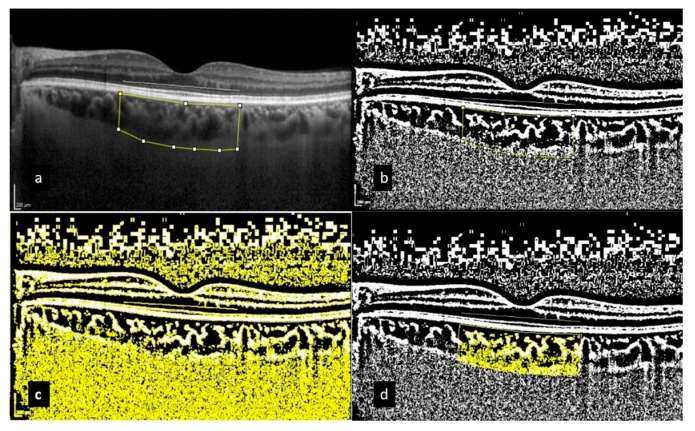
The choroidal vascularity index (CVI) is calculated through the binarization of enhanced-depth imaging (EDI) spectral-domain optical coherence tomography (SD-OCT) images. The process involves (**a**) tracing the choroidal boundaries to determine the total choroidal area (highlighted with yellow lines); (**b**) binarizing the image using Niblack’s auto-local threshold method; (**c**) using the color threshold tool to identify dark pixels, which represent the luminal area; and (**d**) calculating the CVI by dividing the luminal area by the total choroidal area, covering a 3000 µm region.

**Table 1 diagnostics-15-00286-t001:** Demographic findings of the patients with CAE and control group.

	CEA Group *n* = 34	Control Group *n* = 40	*p*
	Mean ± SD or *n* (%)	Mean ± SD or *n* (%)	
**Including eye**			
OD	32 (94.1)	33 (82.5)	0.166
OS	2 (5.9)	7 (17.5)	
**Gender**			
Female	20 (58.8)	24 (60)	0.918
Male	14 (41.2)	16 (40)	
**Age**	61.24 ± 7.9	61.15 ± 8.8	0.644
**Ectatic coronary vessel**			
LCX	6 (17.6)		
LAD	12 (35.3)		
RCA	9 (26.5)		
LMCA	5 (14.7)		
All vessels	2 (5.9)		
**Ectasia morphology**			
Diffuse	9 (26.5)		
Focal	25 (73.5)		
**Hypertension**			
−	18 (52.9)	30 (75)	0.048 *
+	16 (47.1)	10 (25)	
**Diabetes mellitus**			
−	27 (79.4)	34(85)	0.529
+	7 (20.6)	6(15)	
**Smoking**			
−	31 (91.2)	31 (78)	0.112
+	3 (8.8)	9 (22)	
**Hyperlipidemia**			
−	29 (85.3)	39 (99)	0.088
+	5 (14.7)	1 (1)	

*p*: Chi-square test, * significance.

**Table 2 diagnostics-15-00286-t002:** Comparison of subfoveal choroidal thickness and choroidal thickness at 4 cardinal locations between the groups.

	CEA Group *n* = 34		Control Group *n* = 40		*p*
	Min–Max	Mean ± SD	Min–Max	Mean ± SD	
Subfoveal ChT, µm	149–467	297.88 ± 73.2	151–547	336.3 ± 94.4	0.058
Temporal ChT, 500 µm	158–442	290.74 ± 65.3	180–561	328.4 ± 90.2	0.042 *
Temporal ChT, 1500 µm	158–439	280.74 ± 61.5	164–537	306.98 ± 93.5	0.153
Nazal ChT, 500 µm	132–487	283.44 ± 82.9	140–513	329.2 ± 91	0.028 *
Nazal ChT, 1500 µm	100–474	257.62 ± 85	111–527	277.7 ± 90.2	0.33

ChT: choroidal thickness, *p* *: independent samples *t*-test, <0.05.

**Table 3 diagnostics-15-00286-t003:** Comparison of OCT parameters and choroidal vascular index of two groups.

	CEA Group *n* = 34		Control Group *n* = 40		*p*
	Min–Max	Mean ± SD	Min–Max	Mean ± SD	
CMT	190–278	223.44 ± 18.6	190–293	233.43 ± 26.2	0.06
RNFL thickness	83–120	97.91 ± 9.2	75–122	102.1 ± 9.4	0.058
GCC thickness	25–37	31.85 ± 2.6	27–39	33.5 ± 2.7	0.015 *
CVI	0.554–0.696	0.635 ± 0.03	0.627–0.78	0.673 ± 0.03	0.0001 *

CMT: central macula thickness, RNFL: retinal nerve fiber layer, GCC: ganglion cell complex, CVI: choroidal vascular index, *p* *: Mann–Whitney U test, <0.05.

**Table 4 diagnostics-15-00286-t004:** The relationship of systemic risk factors with subfoveal choroidal thickness and choroidal vascular index in patients with CAE.

		Subfoveal Choroidal Thickness	Choroidal Vascular Index
	*n*	Mean ± SD	Mean ± SD
Hypertension			
−	18	308.67 ± 69.1	0.629 ± 0.03
+	16	285.75 ± 77.9	0.641 ± 0.03
*p*		0.37	0.437
Diabetes			
−	27	307.44 ± 70.1	0.636 ± 0.03
+	7	261 ± 78.4	0.63 ± 0.02
*p*		0.25	0.656
Hyperlipidemia			
−	29	302.07 ± 77	0.633 ± 0.03
+	5	273.6 ± 42.3	0.642 ± 0.04
*p*		0.43	0.715
Smoking			
−	31	295.03 ± 75.6	0.634 ± 0.03
+	3	327.33 ± 33	0.647 ± 0.02
*p*		0.474	0.451

*p*: Mann–Whitney U test, independent samples *t*-test.

## Data Availability

The data that support the findings of this study are available from the corresponding author upon reasonable request.

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
