# Peer review of "A Pilot Study Investigating the Relationship Between Choroidal Thickness and Choroidal Vascular Index and Coronary Artery Ectasia"

_diagnostics, 2025, doi:10.3390/diagnostics15030286_

Round 1
Reviewer 1 Report
Comments and Suggestions for Authors
The idea of ​​ correlating the risk of developing coronary heart disease with vascular pathology in other areas, such as the eye, as proposed by this study, is interesting.
However, there are several major problems in this manuscript:
- The very small number of patients included in the study does not give significance to the results obtained
- Today, the correlation between vascular damage in other territories, especially the carotid system and coronary disease is well known. Thus, the measurement of carotid intima thickness and the intima/media index are established as a risk factors for coronary disease.
- Indeed, the study of ocular circulation has been and should continue to be an objective in the evaluation of complications of arterial hypertension, which is also a major coronary risk factor. However, from ocular circulation to coronary events is a very long and unclear path.
- In this context, I consider that the present study does not achieve an objective with possible practical implications and represents only a slightly "extravagant", but not wrong idea.
The references must be updated.
I recommend the authors to continue their research, possibly correlating coronary lesions with concomitant vascular lesions in multiple territories.
Author Response
Dear editor of the Diagnostics Journal,
First of all, we would like to thank all reviewers for their effort and time to review our paper. We are very pleased to see the positive comments from the editor and reviewers. We did try out best to revise our manuscript following their invaluable suggestions and constructive criticisms that will, beyond any doubt, improve the quality of our manuscript. We are eager to see the eventual decision on our manuscript.
Here is my response to the reviewers:
Reviewer 1
The idea of ​​ correlating the risk of developing coronary heart disease with vascular pathology in other areas, such as the eye, as proposed by this study, is interesting.
However, there are several major problems in this manuscript:
Q1- The very small number of patients included in the study does not give significance to the results obtained
A1- Thank you for comments. Although our study included a limited number of patients, the power analysis demonstrated that involving at least 33 patients and healthy individuals were essential to achieve a statistically significant difference of 0.03 between the groups, ensuring a robust power of 0.80. This information was mentioned in the statistical section.
Q2- Today, the correlation between vascular damage in other territories, especially the carotid system and coronary disease is well known. Thus, the measurement of carotid intima thickness and the intima/media index are established as a risk factors for coronary disease.
A2- Thank you for comments. Carotid intima thickness and the intima/media index are widely recognized as significant risk factors for coronary disease. Nevertheless, it is important to note that our study did not include any data regarding carotid intima thickness or the intima/media index. We acknowledged this information in the limitation section of our manuscript.
Q3- Indeed, the study of ocular circulation has been and should continue to be an objective in the evaluation of complications of arterial hypertension, which is also a major coronary risk factor. However, from ocular circulation to coronary events is a very long and unclear path.
A3- Thank you for comments. Of note, this is the first study to pointed out that choroidal thickness and choroidal vascular index can be changed in patients with coronary ectasia. We consider that our pilot study could highlight the need of more prospective studies in this subject.
Q4- In this context, I consider that the present study does not achieve an objective with possible practical implications and represents only a slightly "extravagant", but not wrong idea.
A4- Thank you for comments. We want to underscore that while our study may not yet translate into immediate clinical applications, it highlights a noteworthy association between choroidal thickness, choroidal vascular index, and coronary ectasia. Furthermore, by establishing this crucial connection, we pave the way for further research that will delve deeper into these associations and uncover the clinical implications that can emerge from a better understanding of these vascular changes. This was added in the future direction section of manuscript.
Q5- The references must be updated.
A5- Thank you for comments. References were updated.
Reviewer 2 Report
Comments and Suggestions for Authors
I would like to congratulate the authors on their work. A few recommendations that could further improve their work are outlined below:
a. They can add examples of systemic (diabetes, hypertension, coronary artery disease, smoking) [PMID: 36388726] and ocular conditions (open angle glaucoma and pseudoexfoliation) that have been found to have affected macular choroidal thickness [PMID: 38679155] in lines 60-63.
b. The authors describe the inclusion and exclusion criteria of the study. If they want they can add a figure to describe the flowchart of the included patients and state the number of patients they initially approached, how many did not meet the inclusion/exclusion criteria to arrive at the population of 34 patients with CAE.
c. The authors can include details of the dilatation and the eye drops used in their methods.
d. The authors can include that all measurements were captured on the same time of the day to account for any diurnal variability.
e. The authors can add in their limitations section that other factors that might influence the choroidal thickness such as axial length.
Author Response
Dear editor of the Diagnostics Journal,
First of all, we would like to thank all reviewers for their effort and time to review our paper. We are very pleased to see the positive comments from the editor and reviewers. We did try out best to revise our manuscript following their invaluable suggestions and constructive criticisms that will, beyond any doubt, improve the quality of our manuscript. We are eager to see the eventual decision on our manuscript.
Reviewer 2
I recommend the authors to continue their research, possibly correlating coronary lesions with concomitant vascular lesions in multiple territories. I would like to congratulate the authors on their work. A few recommendations that could further improve their work are outlined below:
Q1- They can add examples of systemic (diabetes, hypertension, coronary artery disease, smoking) [PMID: 36388726] and ocular conditions (open angle glaucoma and pseudoexfoliation) that have been found to have affected macular choroidal thickness [PMID: 38679155] in lines 60-63.
A1- Thank you for comments. Relevant references were added in the text.
Q2- The authors describe the inclusion and exclusion criteria of the study. If they want they can add a figure to describe the flowchart of the included patients and state the number of patients they initially approached, how many did not meet the inclusion/exclusion criteria to arrive at the population of 34 patients with CAE.
A2- Thank you for comments. Flow chart of the study was provided as requested.
Q3- The authors can include details of the dilatation and the eye drops used in their methods.
A3- Thank you for comments. Pupil dilation was achieved through the careful instillation of topical tropicamide 1%, ensuring accuracy and reliability in our results.
Q4- The authors can include that all measurements were captured on the same time of the day to account for any diurnal variability.
A4- Thank you for comments. All measurements were conducted during the early morning hours, specifically between 9:00 AM and 11:00 AM, following pupil dilation to minimize the effects of diurnal rhythm variations.
Q5- The authors can add in their limitations section that other factors that might influence the choroidal thickness such as axial length.
A5- Thank you for comments. In order to ensure the integrity of our study, individuals exhibiting a refractive error greater than ± 3.0 D, along with those having long (≥25 mm) or short (≤22 mm) axial lengths, were excluded from study. Also, other factors such as axial length were added in the limitations section.
Round 2
Reviewer 1 Report
Comments and Suggestions for Authors
The authors have tried to revise the manuscript as best they could, but I maintain my initial opinion that this article cannot bring any real benefit to clinical practice.